# Antibody Response to SARS-CoV-2 among COVID-19 Confirmed Cases and Correlates with Neutralizing Assay in a Subgroup of Patients in Delhi National Capital Region, India

**DOI:** 10.3390/vaccines10081312

**Published:** 2022-08-14

**Authors:** Puneet Misra, Shashi Kant, Randeep Guleria, Sanjay K. Rai, Abhishek Jaiswal, Suprakash Mandal, Guruprasad R. Medigeshi, Mohammad Ahmad, Anisur Rahman, Meenu Sangral, Kapil Yadav, Mohan Bairwa, Partha Haldar, Parveen Kumar

**Affiliations:** 1Centre for Community Medicine, Old OT-Block, All India Institute of Medical Sciences, Ansari Nagar, New Delhi 110029, India; 2Director, All India Institute of Medical Sciences, Ansari Nagar, New Delhi 110029, India; 3Translational Health Science and Technology Institute, Faridabad 121001, India; 4WHO Country Office, New Delhi 110011, India

**Keywords:** SARS-CoV-2, COVID-19, antibody, PRNT, neutralizing antibody

## Abstract

**Background:** The plaque reduction neutralization test (PRNT) is the gold standard to detect the neutralizing capacity of serum antibodies. Neutralizing antibodies confer protection against further infection. The present study measured the antibody level against SARS-CoV-2 among laboratory-confirmed COVID-19 cases and evaluated whether the presence of anti-SARS-CoV-2 antibodies indicates virus neutralizing capacity. **Methods:** One hundred COVID-19 confirmed cases were recruited. Their sociodemographic details and history of COVID-19 vaccination, contact with positive COVID-19 cases, and symptoms were ascertained using a self-developed semi-structured interview schedule. Serum samples of the participants were collected within three months from the date of the positive report of COVID-19. The presence of anti-SARS-CoV-2 antibodies (IgA, IgG and IgM antibodies), receptor binding domain antibodies (anti-RBD), and neutralizing antibodies were measured. **Findings:** Almost all the participants had anti-SARS-CoV-2 antibodies (IgA, IgG and IgM) (99%) and anti-RBD IgG antibodies (97%). However, only 69% had neutralizing antibodies against SARS-CoV-2. Anti-RBD antibody levels were significantly higher among participants having neutralizing antibodies compared with those who did not. **Interpretation:** The present study highlights that the presence of antibodies against SARS-CoV-2, or the presence of anti-RBD antibodies does not necessarily imply the presence of neutralizing antibodies.

## 1. Introduction 

Corona Virus disease (COVID-19) is caused by the virus SARS-CoV-2, which is a single-stranded RNA virus belonging to the genus Betacoronavirus [1]. It emerged in Wuhan, China in December 2019, and was declared a global pandemic by WHO on 11 March 2020.

The SARS-CoV-2 infection causes a wide range of clinical manifestations ranging from cough, fever and malaise to severe pneumonia and acute respiratory distress syndrome [2,3]. Antibody-mediated (humoral immunity) immunity is thought to play a vital role in the protection of both naturally infected and vaccinated people. The SARS-CoV-2 virus induces a classic antibody response in which IgM antibodies appear first followed by IgG antibodies which remain detectable for several months post-symptom onset (PSO), whereas IgM declines by 2–3 weeks PSO [4]. Various serological tests are available to detect these antibodies including enzyme-linked immunosorbent assay (ELISA), lateral flow immunoassay (LFIAs) and chemiluminescent immunoassay (CLIAs) [5]. Serological tests are helpful to identify asymptomatic and previously undiagnosed infections and, thus, are important in epidemiological surveys. Of particular importance are the neutralizing antibodies, which are capable of neutralizing the virus and, thus, provide protection against further infection. The plaque reduction neutralization test (PRNT) is the gold standard to detect the neutralizing capacity of serum antibodies. The receptor-binding protein present in spike protein (S) of the virus interacts with the human acetylcholine esterase-2 (ACE-2) receptor and thus helps the virus enter the host cells [6,7,8]. Blocking the interaction between the S protein receptor-binding domain (RBD) and ACE-2 prevents the entry of the virus and, thus, is the most potent neutralizing epitope offering protection against SARS-CoV-2 [9,10].

The present study was conducted to measure the antibody level against SARS-CoV-2 among COVID-19 positive cases and to evaluate whether the presence of anti-SARS-CoV-2 antibodies indicates virus neutralizing capacity. 

## 2. Methodology

The present study was conducted among 100 participants who were enrolled from 15 March to 31 December 2021 from two sites: one rural site in Ballabgarh, Haryana, and another urban site in Dakshinpuri, New Delhi. Participation was voluntary. All the participants were recruited within 3 months of a positive rapid antigen test report (RAT)/real-time polymerase chain reaction (RT-PCR) report for COVID-19. Participants were enrolled in the study irrespective of their age or current COVID-19 disease status. Participants who refused to give written informed consent, or had contraindication for venipuncture, were excluded from the study. From the consenting participants, we collected information on basic demographic details, exposure history to COVID-19 cases, symptoms suggestive of COVID-19 in the preceding three months and clinical history. 

Blood collection: Trained phlebotomists collected 5 mL of venous blood in plain vials from each participant within three months of testing positive for COVID-19. Serum was separated after centrifugation at the identified local health facility and transported to the respective laboratories for testing. 

Detection of SARS-CoV-2-specific IgG antibodies was performed using an ELISA-based test (WANTAI) as per the specified optical density (OD) cut-off value. Neutralizing antibodies against SARS-CoV-2 were tested using a plaque reduction neutralization test (PRNT) to check antibody titres. Anti-Receptor binding domain (RBD) antibody (IgG) was measured using quantitative RBD ELISA. 

### 2.1. Plaque Reduction Neutralization Test (PRNT)

A PRNT for SARS-CoV-2 on Vero E6 cells was performed to measure the neutralizing antibodiesas reported previously [11]. PRNT_50_ was reported in titres. PRNT_50_ titre > 20 was reported as positive, and PRNT_50_ titre of 20 or less was reported as negative. The PRNT for SARS-CoV-2 had a measurement uncertainty of ±19.14 at 936 PRNT_50_ titres of serum.

The basic design of the PRNT assay allows virus-antibody interaction to occur in a microtiter plate, and then a virus-antibody mixture was added to virus-susceptible cells. Vero E6 cells from ECACC (Salisbury, UK, Cat no. 85020206) maintained in complete Dulbecco’s modified Eagle medium (HiMedia, India. Cat No. -AL007A), which contains 10% heat inactivated fetal bovine serum (Gibco, USA. Cat no. 16140-071), 1× Penicillin Streptomycin (HiMedia, Mumbai, India, Cat No.-A001) and 1× non-essential amino acids solution (NEAA) (Gibco, Waltham, MA, USA, Cat no. 1140050), were seeded at 150,000 cells per well in a 24-well plate. The serum samples were heat-inactivated at 56 °C for 30 min and were serially diluted from 1:10 to 1:1280 to make a final volume of 75 µL. To this, 75 µL (adjusted to provide ~20–60 plaques/well) of virus suspension (B.6. ancestral strain GenBank accession no: MW422884) prepared in assay diluent which contains Dulbecco’s modified Eagle medium (HiMedia, Mumbai, India, Cat No. -AL007A), which contains 2% heat inactivated fetal bovine serum (Gibco, Waltham, MA, USA, Cat no. 16140-071), 1× Penicillin Streptomycin (HiMedia, Mumbai, India, Cat No.-A001) and 1× non-essential amino acids solution (NEAA) (Gibco, Waltham, MA, USA, Cat no. 1140050), was added. Hence, the final dilution of the serum range was from 1: 20 to 1: 2560 and the virus-serum mixture was kept for 1 h at 37 °C in a 5% CO_2_ incubator for virus neutralization. The virus-serum mixtures were then added to a confluent monolayer of Vero E6 cells and incubated for 1 h at 37 °C in a 5% CO_2_ incubator for virus adsorption. After 1 h, the viral inoculum was removed and overlaid with 0.5% Carboxymethylcellulose (CMC) (Sigma, St. Louis, MO, USA, Cat no: C4888) assay diluent as described above. At 48 h post-infection, the cells were fixed with 3.7% formaldehyde solution (Merck, Kenilworth, IL, USA. Cat no: 1.94989.0521) and stained with 1× crystal violet solution (Sigma, St. Louis, MO, USA, Cat no: C0775). Plaques were counted manually. PRNT50 values were calculated with a 4-parameter logistic regression using GraphPad Prism 9.0 software.

The measurement of uncertainty for PRNT was calculated based on the ISO 17025:2017 guideline. The uncertainty was directly related to the measurement parameter, range of the measurement, the equipment or measurement process being used (affecting precision), and the standards available with associated uncertainties. The uncertainty in the analytical result was taken into account when assessing compliance. It is the expression of the statistical dispersion of the values attributed to a measured quantity and based on various factors such as micropipettes and other equipment as well as variation in data points. 

### 2.2. QRBD

Quantitative enzyme-linked immunosorbent assay (ELISA) was used to estimate serum IgG antibodies binding to the receptor-binding domain of SARS-CoV-2 Spike protein as reported previously [12]. The test reported the anti-RBD IgG antibodies in ELU/mL. QRBD ≥ 12.0 ELU/mL was reported as positive, and between 8.0 and <12.0 ELU/mL was reported as equivocal. QRBD < 8.0 ELU/mL was reported as negative.

The SARS-CoV-2 RBD IgG ELISA was performed using a two-step incubation immuno-assay. 

From BEI resources (US, Cat no: NR-52422), mammalian expression vector pcDNA™3.1 (+) comprising the codon-optimized gene sequence of the receptor-binding domain (RBD, amino acids 328–531) of spike (S) glycoprotein from SARS-CoV-2, Wuhan-Hu-1 (GenBank: MN908947) with an N-terminal mu-phosphatase signal sequence and C-terminal octa-histidine tag was obtained. The RBD protein was expressed and purified in its fully glycosylated form through the mammalian expression system (Expi293F cells, Gibco, USA. Cat no: A14527). The recombinant RBD antigen (2 µg/mL) of SARS-CoV-2 spike protein in PBS (phosphate-buffered saline) was coated onto 96-well MaxiSorp ELISA plates (NUNC, New York, NY, USA, Cat no: 442404) (50 µL/well) and incubated at 4 °C for 18–22 h. The antigen-coated plates were washed with wash buffer (1× PBST) (phosphate-buffered saline with 0.1% Tween 20 (Sigma-Aldrich, St. Louis, Mo, USA, Cat no: P13790) and incubated by adding 200 µL of blocking buffer (3% non-fat dry milk powder (Bio-rad, Hercules, FL, USA, Cat no: 1706404) in PBST) and incubated at RT (23 ± 2 °C) for 1 h. The serum samples were inactivated with Triton X-100 (Sigma, Louis, MO, USA. Cat no: T8787 and) and were diluted at 1:50 or 1:500 in blocking buffer, and 100 µL of diluted serum was added to each well in two replicates and incubated at RT (23 ± 2 °C) for 30–40 min followed by washes and secondary antibody (Jackson Immuno Research, USA. Cat no: 109-035-170). After removal of nonspecific binding, an HRP substrate solution containing 3,3′,5,5′-Tetramethylbenzidine (TMB) (BD, Canada. Cat no: 555214) was added, resulting in the formation of a blue colour. The colour reaction was stopped by 1M H_2_SO4 (Merck, India. Cat no: 1.93400.0521), which transforms the colour of the solution from blue to yellow. The intensity of the colour was quantified by measuring absorbance in a microplate reader at 450 nm with a reference wavelength of 630 nm. The colour intensity was directly proportional to the amount of anti-RBD antibodies captured inside the wells. 

NIBSC (National Institute for Biological Standards and Control) 20/130 research reagent with an assigned concentration of 502 ELISA Units/mL (ELU/mL) was used as standard reference material. In-house standard serum, obtained from convalescent COVID-19 patients, was calibrated against the WHO reference reagent and was used as a secondary standard. The lower limit of detection for the assay was 8 ELU/mL. The antibody concentrations were calculated for each sample dilution by interpolation of the OD values on the 4-parameter logistic (4-PL) standard curve from positive control and adjusted according to their corresponding dilution factor using Gen5 software (BioTek Instruments, Winooski, VT, USA). This assay has been validated in-house and accredited under the ISO 17025:2017 standard.

### 2.3. WANTAI SARS-CoV-2 Antibody ELISA

An enzyme-linked immunosorbent assay (ELISA) was performed for the qualitative detection of total antibodies to the SARS-CoV-2 virus in human serum or plasma specimens (anti-SARS-CoV-2 IgA, IgG and IgM antibodies). The kit is intended for screening of patients suspected of infection with the SARS-CoV-2 virus, and as an aid in the diagnosis of the coronavirus disease 2019 (COVID-19). Specimens with OD ≥ 0.19 were considered positive, and <0.19 negative. 

WANTAI SARS-CoV-2 Ab ELISA was performed using a two-step incubation antigen “sandwich” enzyme immunoassay kit (Beijing Wantai Biological Pharmacy Enterprise, Beijing, China. Cat no: WS-1096). The test was performed according to the instructions of the manufacturer. The kit used polystyrene microwell strips pre-coated with recombinant SARS-CoV-2 antigen. 

The patient’s serum specimen (100 μL) was added, and during the first incubation (30 min at 37 °C), the specific SARS-CoV-2 antibodies were captured inside the wells if present. The microwells were then washed (five times) to remove unbound serum proteins. Second, recombinant SARS-CoV-2 antigen conjugated to the enzyme horseradish peroxidase (HRP-Conjugate, 100 μL) was added, and during the second incubation (30 min at 37 °C), the conjugated antigen bonded to the captured antibody inside the wells. The microwells were then washed (five times) to remove unbound conjugate, and Chromogen solutions (50 μL of Chromogen A and Chromogen B) were added to the wells. The plate was incubated at 37 °C for 15 min avoiding light. In wells containing the antigen-antibody-antigen (HRP) “sandwich” immune-complex, the colourless Chromogens are hydrolyzed by the bound HRP conjugate to a blue-coloured product. The blue colour turns yellow after the reaction is stopped by adding 50 μL of stop solution to each well (sulfuric acid). The degree of colour intensity was measured and represented the amount of antibody captured inside the wells, and in the specimen. Wells containing specimens negative for SARS-CoV-2 antibodies remained colourless. Specimens with an absorbance to cut-off ratio of ≥1.0 were considered positive.

### 2.4. Statistical Analysis

Categorical variables are reported as frequencies and percentages. The normality of continuous variables was tested using the Shapiro–Wilk test. Continuous variables are reported as medians with an interquartile range. The Wilcoxon rank-sum test was applied to test the statistical significance of continuous variables. For testing the correlation of categorical variables, the Cramer V was calculated and for non-normal continuous variables, Spearman’s Rank correlation coefficients were calculated. 

## 3. Results

The mean (S.D.) age of the participant was 37.0 (13.5) years, and the ages ranged from 14 years to 72 years. The majority of the participants were male (64%). The majority of the participants (63%) had a history of fever, followed by cough (42%), sore throat (35%), and loss of taste sensation (24%). Seventy-four participants had at least one symptom, and the remaining 26 were asymptomatic. 

All the participants were laboratory-confirmed cases of COVID-19, and the majority (80%) were RTPCR positive, and the rest (20%) were RAT positive. 

Only 22 participants had received vaccination against COVID-19. Among these 12 participants had received two doses, while others (10) had received only a single dose. The median (IQR) gap (in days) between receiving the first dose of COVID-19 vaccine and sample collection was 80 (66–127) days; similarly, the gap between second dose and sample collection was 86 (62–108) days.

Thirty-one participants had a PRNT_50_ titre of less than 20, considered negative for neutralizing antibodies. Sixty-nine participants had neutralizing antibodies (PRNT_50_ titre ≥ 20). Among the participants who had neutralizing antibodies (PRNT_50_ titre ≥ 20), 49 (71%) participants had PRNT_50_ titre ≥ 80, 37 (54%) had PRNT_50_ titre ≥ 160, and 26 (38%) had PRNT_50_ titre ≥ 320 (Figure 1) (Categories are not mutually exclusive). 

Almost all participants (97.0%) were positive for anti-RBD antibody (Serum IgG against receptor binding domain of COVID-19, determined by QRBD) (≥12.0 ELU/mL), and three (3.0%) participants had equivocal results for QRBD (>7.99 to <12.0 ELU/mL). None of the participants were QRBD negative. 

Among the 97 participants who were positive for anti-RBD antibody, 69 (71.1%) had neutralizing antibodies (PRNT_50_ titre ≥ 20). All the three participants with an equivocal result for the QRBD had a PRNT_50_ titre of less than 20. 

Almost all participants (99%) were positive for total anti-SARS-CoV-2 antibodies (IgA, IgG, and IgM) (≥0.19 optical density (OD) in the WANTAI assay). 

The Cramer V (correlation coefficient) for presence or absence anti-RBD antibody with presence/absence of neutralization antibody (PRNT_50_ ≥ 20 Presence, PRNT_50_ < 20 absence) was 0.26 (*p*-value = 0.028). However, the Spearman’s ranks correlation coefficient for anti-RBD antibody value (ELU/mL) with the neutralization antibody titre was 0.78 (Spearman’s rho, *p*-value < 0.0001). 

The distribution of the PRNT_50_ titre and anti-RBD antibody levels showed a non-normal distribution (Shapiro–Wilk test, *p* < 0.001, for both the variables). The median (IQR) value for the PRNT_50_ titre was 71 (19, 415.5). Similarly, the median (IQR) value for the QRBD (ELU/mL) level was 202 (60, 627.6).

Figure 2 and Figure 3 show the distribution of the PRNT_50_ (titre), and QRBD (ELU/mL) for the 100 participants. Figure 4 shows the scatter plot of anti-RBD antibodies (ELU/mL) (QRBD) among the participants with neutralizing antibody titre (PRNT). 

Table 1 shows the median (IQR) PRNT_50_ titre for different sociodemographic/clinical variables. The median (IQR) PRNT_50_ titre was significantly higher among those who received at least one dose of the COVID-19 vaccine compared with no vaccination [590 (115, 1204) vs. 45 (19, 197), Wilcoxon rank-sum test, *p*-value = 0.01]. 

The median (IQR) PRNT_50_ titre among those who received two doses was higher compared to those who received a single dose, though the difference was not statistically significant [344.5 (972.5, 2787.5) vs. 119 (65, 770), Wilcoxon rank-sum test, *p*-value = 0.069].

Similarly, the median (IQR) PRNT_50_ titre was significantly higher among participants who were working as health care workers compared to those who were not [861 (410, 2922) vs. 58 (19, 236), Wilcoxon rank-sum test, *p*-value < 0.01] (Table 1). 

The median (IQR) anti-RBD antibody level was significantly higher among residents of urban areas compared with the rural area [437.6 (141.9, 1183.4) vs. 192.1 (55.4, 589.8), Wilcoxon rank-sum test, *p*-value = 0.04]. Those who had taken at least one dose of the COVID-19 vaccine had a significantly higher median (IQR) anti-RBD antibody levels compared to unvaccinated [718.1 (441.3, 1415.9) vs. 131 (52.3, 372.3), Wilcoxon rank-sum test, *p*-value < 0.01]. Participants who had a history of contact with COVID-19 positive cases had a significantly higher median (IQR) QRBD titre of 348.2 (126.8, 1094.8) compared with those who had no history of contact with COVID positive cases (128.2 (48.7, 414.8)) (Wilcoxon rank-sum test, *p*-value < 0.01). Participants who were working as health workers also had a significantly higher median (IQR) QRBD titre compared with those who were not health workers [798.4 (441.3, 1415.9) vs. 184.7 (55.6, 560.7), Wilcoxon rank-sum test, *p*-value < 0.01].

A similar distribution as for the PRNT titre was seen for anti-RBD antibody levels to sociodemographic/clinical variables, except for two variables: (a) History of contact: the median PRNT_50_ titre was higher amongst those exposed, whereas the median anti-RBD antibody levels were higher among those not exposed; and (b) Presence/absence of fever as a symptom: the median PRNT_50_ titre was higher among those who had no fever, whereas the median anti-RBD antibody levels were higher among those reporting fevers. Among the participants with anti-RBD antibodies: the anti-RBD antibody levels were significantly higher for participants who had neutralizing antibodies compared with those who did not (Wilcoxon rank-sum test: *p*-value < 0.001). Overall, the anti-RBD antibody levels were also significantly higher among participants who had neutralizing antibodies compared with participants who did not (Wilcoxon rank-sum test: *p*-value < 0.001) (Table 2), (Figure 5). Figure 5 also clearly shows that the participants who had anti-RBD antibodies higher than 300 ELU/mL all had a PRNT_50_ titre in the positive range. 

## 4. Discussion

The present study was conducted among one hundred laboratory confirmed COVID-19 positive cases. All the participants were tested by PRNT, WANTAI and QRBD for COVID-19 neutralizing antibodies, total antibodies (IgA, IgG, and IgM) against COVID-19, and anti-RBD IgG antibodies for COVID-19 respectively. 

In the present study, although almost all the participants had anti-RBD IgG antibodies and anti-COVID-19 antibodies, the same was not true for the presence of neutralizing antibodies. Therefore, just the presence of anti-SARS-CoV-2 antibodies does not mean that the person has neutralizing antibody titre and is thereby protected against the virus. 

In the study by Deshpande et al. [13], among 343 participants, 71.9% developed neutralizing antibodies to SARS-CoV-2. Among the 28.1% (*n* = 25) participants who failed to develop neutralizing antibodies, eleven participants were positive by anti-SARS-CoV-2 IgG ELISA. The participants in their study differed from our study. We included only laboratory-confirmed COVID-19 cases; however, their study sample consisted of a mixed sample (89 positive, 58 negatives for SARS-CoV-2 and 17 cross-reactive and 179 serums from healthy participants). They also reported PRNT_90_ instead of PRNT_50_ as was the case in this study. The difference observed, therefore, could be due to differences in methods. 

Lau et al. [14] reported that 99.1% of the participants had neutralizing antibodies at 90 days after symptoms/detection of infection in serum samples from 195 RTPCR positive cases of COVID-19. The difference could be due to the difference in the disease spectrum of the recruited patients. The study by Lau et al. had only 31 asymptomatic cases (15%), whereas in the present study, 26% of individuals were asymptomatic. 

Spearman’s rho for correlation between anti-RBD antibodies and neutralizing antibodies was high, which is similar to reports from the previous studies [15,16], indicating that as the anti-RBD antibody values increase, so do the neutralizing antibody titres. However, the correlation between the presence/absence of anti-RBD antibodies and neutralizing antibodies as a dichotomous variable was low (as reported by low Cramer’s V), though this could be because of the lower COVID-19 vaccination rate in the present study, which is known to affect the correlation between the anti-RBD antibodies and neutralizing antibodies [16,17]. 

However, the present study also shows that there is a need to increase the cut-off point of the anti-RBD antibody levels which then can act as a proxy indicator for the presence of neutralizing antibodies, as all the participants with anti-RBD antibodies > 300 ELU/mL had neutralizing antibodies (Figure 5). 

It is also well known that the total antibodies and neutralizing antibodies decline over time, and in the absence of an adequate antibody level, cell-mediated immunity has been found to have protection. Since cell-mediated immunity in COVID-19 is well established, there might be the possibility of cell-mediated immunity in the participants not having neutralizing antibodies. 

Significantly higher neutralizing antibody titres and anti-RBD antibodies among vaccinated participants signify that vaccination protects against COVID-19. This finding is in agreement with the previous studies [16,17]. 

Significantly higher neutralizing titres and anti-RBD antibodies among the health care workers might be due to higher vaccination rates against COVID-19 among them as well as repeated exposure to SARS-CoV-2. 

Similarly, one of the reasons for higher anti-RBD antibodies among participants who were urban residents and who had a history of contact with COVID-19 cases could be repeated exposure. Additionally, higher population density in urban areas compared with rural areas could lead to a higher probability of repeated exposure to SARS-CoV-2. 

Strengths: Only laboratory-confirmed cases of COVID-19 were recruited in this study. We measured neutralizing antibodies through the PRNT assay, which is considered the gold standard. Additionally, all the sera samples were tested for anti-RBD antibodies and total antibodies. Standard kits and protocols were followed for all the assays. All the sera samples were taken within 3 months of positive RT-PCR/RAT tests. We have also compared the titre of neutralizing antibodies to vaccination status, and dose of vaccine. Furthermore, those participants who were vaccinated were sampled within 6 months of the first dose of vaccination. 

Limitations: The symptoms and history of contact were self-reported, making them vulnerable to recall error. Due to resource limitations, only 100 participants could be included in the study. Immunity in COVID-19 could be due to both cellular and humoral immunity. As we have only tested for humoral immunity, this could be another limitation of the study. Additionally, as almost one-third of the COVID-19 infected participants were lacking neutralizing antibodies, the possibility of other means of protection such as cytotoxic-T cells could not be explored through the present study. Finally, in the present study, the COVID-19 vaccination rate was lower, which could affect the finding observed. 

## 5. Conclusions

Almost all the participants had anti-SARS-CoV-2 antibodies (IgG and IgM) and anti-RBD IgG antibodies. However, only 69% had neutralizing antibodies against SARS-CoV-2. The proportion of participants with higher titres of neutralizing antibodies was even lower at almost 50%. The present study highlights that the presence of antibodies against SARS-CoV-2, or the presence of anti-RBD antibodies doesn’t necessarily imply the presence of neutralizing antibodies. 

## Figures and Tables

**Figure 1 vaccines-10-01312-f001:**
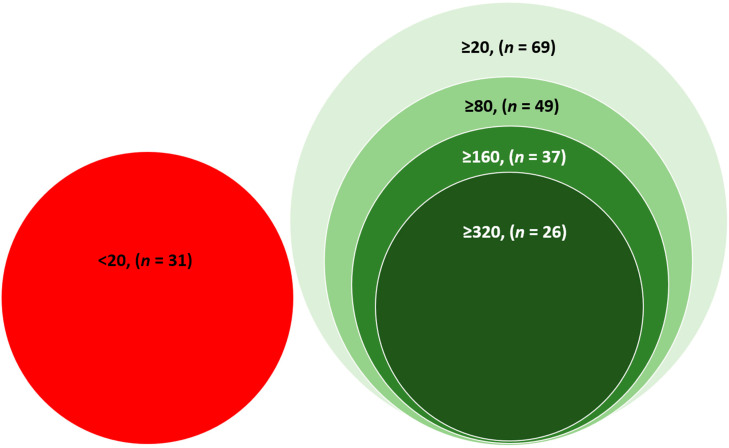
Venn diagram showing the PRNT_50_ titre of the study participants (<20 titres is considered negative, ≥20 is considered positive for the presence of neutralizing antibodies).

**Figure 2 vaccines-10-01312-f002:**
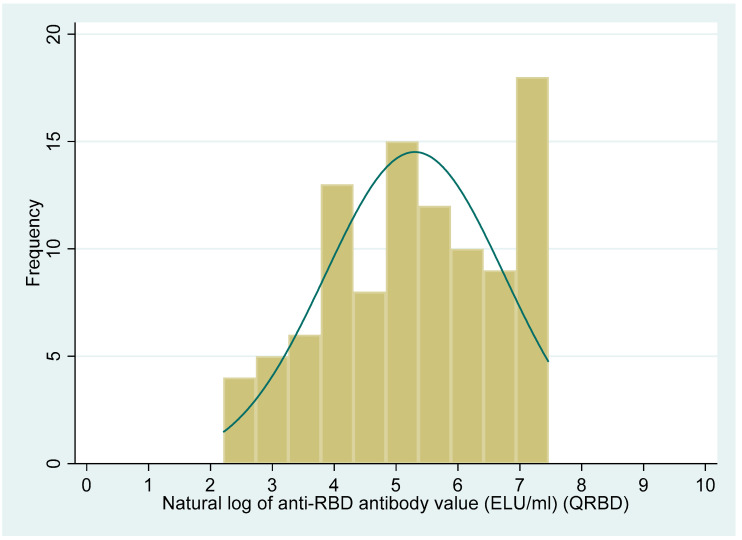
Distribution of anti-RBD antibodies (ELU/mL) among study participants (the scale is a natural logarithm for anti-RBD antibodies).

**Figure 3 vaccines-10-01312-f003:**
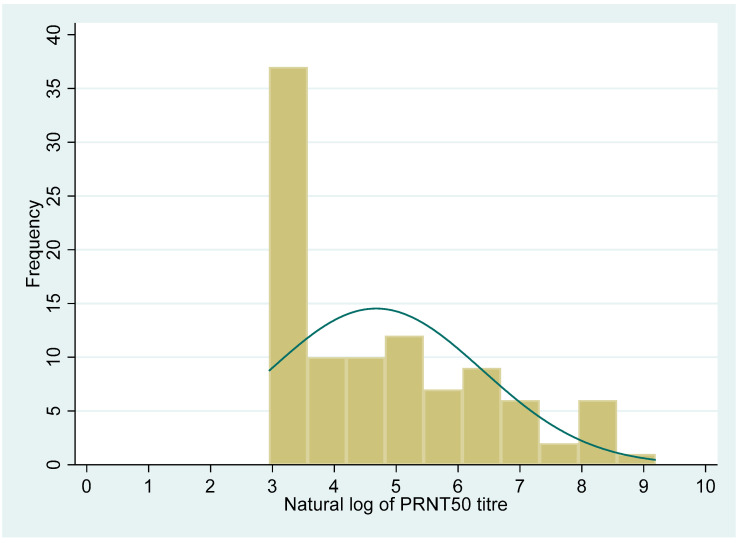
Distribution of neutralizing antibody titres (PRNT_50_) among the participants (the scale is a natural logarithm for PRNT_50_ titre).

**Figure 4 vaccines-10-01312-f004:**
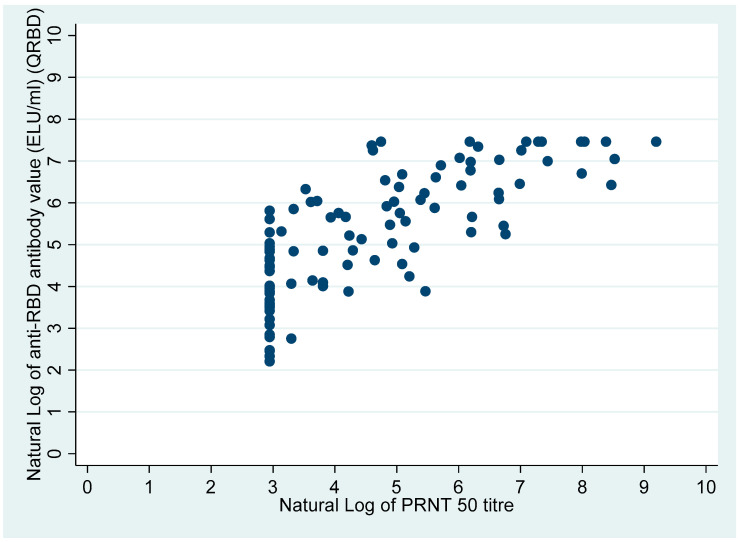
Scatter plot of anti-RBD antibodies (ELU/mL) among participants with neutralizing antibody titres (The scales are natural logarithms).

**Figure 5 vaccines-10-01312-f005:**
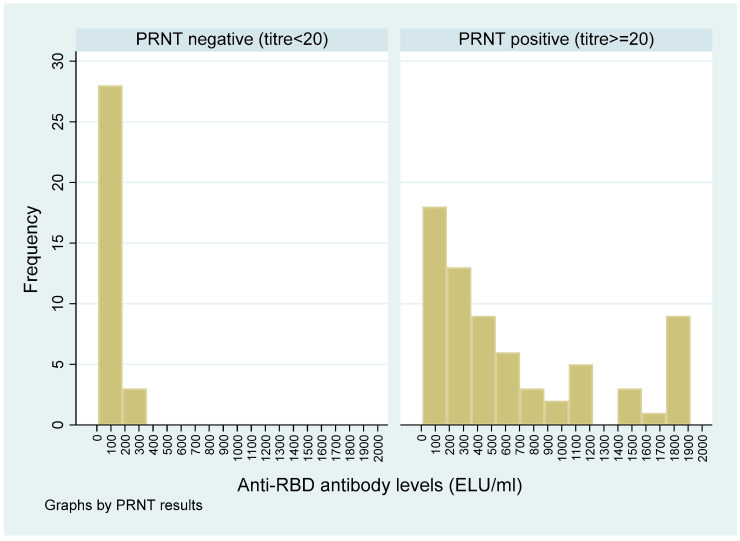
Histogram showing anti-RBD antibodies (ELU/mL) among the participants with respect to PRNT_50_ < 20 and PRNT_50_ ≥ 20.

**Table 1 vaccines-10-01312-t001:** Distribution of PRNT titre by sociodemographic and clinical variables.

Variable			PRNT Titre			
		N	Median	Q1	Q3	*p*-Value *
PRNT		100	71	19	415.5	-
COVID-19 confirmatory test	RTPCR (+)	80	68	19	288.5	0.65
	RAT (+)	20	100	19	633.5	
Sex	Male	64	108	19	497.5	0.21
	Female	36	54.50	19	238	
Residence	Rural	82	66	19	274	0.07
	Urban	18	200	34	861	
COVID-19 vaccination status	No	78	45	19	197	0.01
	Yes	22	590	115	1204	
Seek medical attention	No	87	69	19	279	0.39
	Yes	13	126	19	2952	
Miss school or work	No	85	69	19	410	0.98
	Yes	15	84	19	494	
Hospitalized	No	90	68.50	19	279	0.35
	Yes	10	286	19	770	
History of contact (+)	No	49	67	19	553	0.9
	Unknown	13	84	27	197	
	Yes	38	102.50	19	232	
Wear face mask	Yes	100	71	19	415.5	-
Health worker	Yes	9	861	410	2922	<0.01
	No	91	58	19	236	
Asymptomatic	No	74	71	19	303	0.68
	Yes	26	76	27	494	
Fever	No	37	104	27	483	0.55
	Yes	63	67	19	410	
Sore throat	No	65	68	19	236	0.75
	Yes	35	126	19	501	
Cough	No	57	58	19	236	0.46
	Yes	42	135.50	19	421	
Shortness of breath	No	87	69	19	303	0.48
	Yes	13	126	19	775	
Loss of smell	No	84	91.50	19	455	0.25
	Yes	16	32	19	283	
Loss of taste	No	76	91.50	19	452	0.28
	Yes	24	34	19	286	

* Wilcoxon rank-sum test/Kruskal–Wallis test.

**Table 2 vaccines-10-01312-t002:** Distribution of anti-RBD antibody levels by neutralizing antibody titre status.

Anti-RBD Antibody Status	PRNT_50_ Titre Status				
	Negative (PRNT_50_ < 20)		Positive (PRNT_50_ ≥ 20)		Wilcoxon Rank-Sum Test (*p*-Value) PRNT(+) vs. PRNT (-)
	Frequency (%)	Anti-RBD antibody Levels (ELU/mL) Median (IQR)	Frequency (%)	Anti-RBD antibody Levels s (ELU/mL) Median (IQR)	
Positive	28 (28%)	53.8 (34.0, 116.75)	69 (69%)	414.8 (169.2, 1073.7)	<0.001
Equivocal	3 (3%)	11.7 (9.1, 11.9)	0 (0)	-	-
Total	31 (31%)	48.7 (25.0, 107.4)	69 (69%)	414.8 (169.2, 1073.7)	<0.001

## Data Availability

The data presented in this study are available on request from the corresponding author with a signed data access agreement. The data are not publicly available due to privacy and ethical issues.

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
