# Peer review of "Antibody Response to SARS-CoV-2 among COVID-19 Confirmed Cases and Correlates with Neutralizing Assay in a Subgroup of Patients in Delhi National Capital Region, India"

_vaccines, 2022, doi:10.3390/vaccines10081312_

Round 1

Reviewer 1 Report

This manuscript reported by Misra et al. presents their statistical data on the level of anti-SARS-CoV-2 antibodies generated among one hundred COVID-19 infected individuals. The participants were enrolled from patients in Delhi National Capital Region, India. Their serum samples were collected within three months from the date when the positive COVID was reported. This study aims to find whether there is correlation between the presence of antibodies against SARS-CoV-2 and neutralizing antibodies. By using the published methods, the authors measured the level of antibodies, including anti-SARS-CoV-2 antibodies (i.e., IgA, IgG, and IgM antibodies), receptor binding domain antibodies (anti-RBD), and neutralizing antibodies in the serum or plasma specimens. They found that 99% of the participants had anti-SARS-CoV2 antibodies and 97% of them had anti-RBD IgG antibodies. Nevertheless, only 69% had neutralizing antibodies against SARS-CoV-2.

According to the previous papers published by other research groups, Musser et al. and Guiomar et al. (see refs 13 and 14, respectively, in this manuscript) found a good correlation between the anti-RBD antibodies and neutralizing antibodies. However, Misra et al. revealed a different picture in this manuscript that the presence of anti-RBD antibodies does not necessarily imply the presence of neutralizing antibodies. Their approach is typical but the discussion of the statistical results is not profound enough regarding to their view deviated from the published papers. The major shortcomings of this manuscript are summarized as follows.

1)    Several variables listed in Table 1 are not scientifically sufficient or concrete to evaluate or quantify. “History of contact (+)” and “wear face mask” are variables that are difficult to assess. For example, each individual may have one or several contacts with infected individuals. Face masks could be worn differently person by person, such as various duration for each day. In addition, “any symptom” is too generalized and broad to be included as a variable. On the other hand, the key variables like how many jabs of vaccine have been received by the sample individuals and whether the individuals have been re-infected are missing in this study. Those variables are crucial and countable and should be taken into consideration in the statistical studies.

2)    The Venn diagram shown in Figure 1 for the positive and negative presence of neutralizing antibodies is not proportional in terms of the relation between number of the samples and sizes of the circular area. For example, the circular area of n = 69 for the positive is 9 times larger than that of n = 31 for the negative. Moreover, the size of n = 26 is bigger than that of n = 31. Though a Venn diagram is generally used for showing the logical relation between sets, the scale of proportion should be rational. Such disproportion may cause some misleading.

3)    The strength described on p. 10 is a standard protocol used for statistical studies and antibodies measurement. The authors did not discuss in depth about their approaches and interpret their results. Moreover, the description about the limitations is too simple with only one sentence.

Overall, this manuscript is not convincing to support their new findings. Thus it is not recommended for its publication in Vaccines at its current form.

Author Response

Dear Sir/Ma'am, 

We have made the necessary changes in the manuscript as suggested. We are attaching the reply to your comments. 

Thank you 

Reviewer 2 Report

This manuscript reports on the levels of anti-SARS-CoV2 neutralizing antibodies, quantitative assessment of specific IgG and all isotypes in a sandwich method, in 100 patients having suffered from COVID19. This is an already well covered field and the authors merely confirm that seropositivity is not synonym to neutralization.

In material and methods, more detail should be provided about the source of reagents (name of test or compound, virus strain, manufacturers, instruments…) while the reference of SOP is irrelevant. The methodology should also be described more thoroughly. It is generally admitted that such descriptions should allow people in another laboratory to reproduce the tests, i.e. quantities, incubation times…. This is not the case here. Similarly, it is not known how the uncertainty of the PRNT was assessed.

The subjects tested are reported as having suffered from COVID19 and are dubbed « laboratory confirmed ». What were the proportions of RAT and PCR-positive tests ? Do these two groups differ in antibody results ?

The authors mention that their data do not obey to normal distribution. Only medians and IQR should therefore be shown. Between tests correlations should be performed more rigorously showing the numbers of +/+, +/- and -/+ cases.

Figures 2 to 5 are confusing giving the impression that most subjects are devoid of antibodies. A logarithmic scale should be used to accommodate the high levels.

The paragraph looking for differences in titers depending on a series of criteria is redundant within itself and with table 1. Moreover, only statistically significant differences (p<0.05) warrant being commented. Another confusing point is the fact that 22 subjects were vaccinated, although no notion of delay since COVID or vaccination is given for them.

The discussion is highly redundant in repeating the results section. Because the authors tested humoral immunity in various ways, they could emphasize this point and also mention cellular immunity. Indeed the lack of neutralizing antibodies in previously infected yet healed persons suggests that other means of protection such as cytotoxic T-cells are active. There begins to be some literature about this point in COVID.

Minor

Two sentences are duplicated in the methods section.

In the QRBD test, it is more likely that the antibodies recognize the antigen than the reverse.

The reference style is not homogeneous.

The manuscript should be reviewed for proper English language usage.

Author Response

Dear Sir/Ma'am, 

We have made the suggested changes in the manuscript as advised. We are also attaching the reply to your comments. 

Thank you 

Round 2

Reviewer 1 Report

The authors have made extensive revisions and improvements on their original manuscript. They also have answered most of the questions and concerns raised by the reviewer.  However, there are still some queries which need more explanation from the authors so that the conclusion would not be screwed. The major ones are particularly related to the Discussion section. 

1.    The 8th paragraph of the Discussion on p. 10 ­-- It is stated by the authors that “Significantly higher neutralizing titre and anti-RBD antibody among the health care workers might be due to repeated exposure to SARS-CoV2 among them.” Health workers generally have a higher percentage of vaccination compared to non-health workers. Thus besides the reason of more chances of exposure to SARS-CoV-2, the higher counts of health workers are very likely due to vaccination?  

2.    The 7th paragraph of the Discussion on p. 10 – It is stated by the authors that “Significantly higher neutralizing antibody titre and anti-RBD antibodies among vaccinated participants signify that vaccination protects against COVID-19. This finding is in agreement with the previous studies. [16,17]” Thus when the individuals are vaccinated there is a good (high) correlation between the anti-RBD antibodies and neutralizing antibodies (also see refs. 16 and 17 in the revised manuscript).This point has to be clarified by the authors and described in the Limitations.

3.    In this manuscript, the percentage of vaccination is merely 22% (less than 50%). As vaccination plays a critical on the correlation between the anti-RBD antibodies and neutralizing antibodies, the difference between this manuscript and the published papers reported by other research groups may come from the low percentage of vaccination in this study?

Thus I still do not recommend this manuscript to be accepted for publication at its present form for Vaccines.

Author Response

Dear Reviewer,

Kindly find the attached reply. We have incorporated the suggested changes

Reviewer 2 Report

The manuscript has been improved according to this reviewer’s comments but some detail remains to be amended.

The introduction is much better.

The methods section is improved but:

-         Not all manufacturers are mentioned

-         Their city and state/country that should be mentioned at first appearance is lacking in most cases

-         Catalog numbers are not necessary

The last two sentences before Table 1 lack verbs and should be rephrased

Log scales have greatly improved the figures.

The result and discussion sections are far easier to read and more informative.

The Vancouver style is year;volume:page-page

English usage is better except for the two sentences mentioned. Moreover, in the abstract, use does not rather that doesn’t

Author Response

(The authors gave the same response as above.)
